# Compensation or Displacement of Physical Activity in Children and Adolescents: A Systematic Review of Empirical Studies

**DOI:** 10.3390/children9030351

**Published:** 2022-03-03

**Authors:** Franziska Beck, Florian A. Engel, Anne Kerstin Reimers

**Affiliations:** 1Department of Sport Science and Sport, Friedrich-Alexander-Universität Erlangen-Nürnberg, Gebbertstraße 123b, 91052 Erlangen, Germany; anne.reimers@fau.de; 2Institute of Sport Science, Julius-Maximilians-University Würzburg, Judenbühlweg 11, 97082 Würzburg, Germany; florian.engel@uni-wuerzburg.de

**Keywords:** compensation, displacement, physical activity, children, adolescents

## Abstract

Regular physical activity during childhood and adolescence is associated with health benefits. Consequently, numerous health promotion programs for children and adolescents emphasize the enhancement of physical activity. However, the ActivityStat hypothesis states that increases in physical activity in one domain are compensated for by decreasing physical activity in another domain. Currently, little is known about how physical activity varies in children and adolescents within intervals of one day or multiple days. This systematic review provides an overview of studies that analyzed changes in (overall) physical activity, which were assessed with objective measurements, or compensatory mechanisms caused by increases or decreases in physical activity in a specific domain in children and adolescents. A systematic search of electronic databases (PubMed, Scopus, Web of Science, SportDiscus) was performed with a priori defined inclusion criteria. Two independent researchers screened the literature and identified and rated the methodological quality of the studies. A total of 77 peer-reviewed articles were included that analyzed changes in overall physical activity with multiple methodological approaches resulting in compensation or displacement. Of 40,829 participants, 16,265 indicated compensation associated with physical activity. Subgroup analyses separated by study design, participants, measurement instrument, physical activity context, and intervention duration also showed mixed results toward an indication of compensation. Quality assessment of the included studies revealed that they were of high quality (mean = 0.866). This review provides inconclusive results about compensation in relation to physical activity. A trend toward increased compensation in interventional studies and in interventions of longer duration have been observed.

## 1. Introduction

### 1.1. Health Benefits of Physical Activity

Regular physical activity (PA) during childhood and adolescence is associated with numerous health benefits [1]. As a result of regular PA, physical fitness in children increases, and is associated with improvements in cardiovascular [2] and cardiometabolic [1,2] health, as well as with a reduction in obesity risk [3,4,5]. Additionally, PA is associated with better mental health in children and adolescents [6,7].

In line with these findings, the World Health Organization (WHO) [8,9] developed PA recommendations for children and adolescents. Briefly, it is recommended for children 5 to 17 years of age to accumulate at least 60 min of moderate-to-vigorous PA (MVPA) daily [9]. However, recent analyses have demonstrated that PA levels of children and adolescents in many regions worldwide are not meeting the WHO recommendations [10,11,12].

### 1.2. ActivityStat Hypothesis and Compensatory Mechanisms

PA promotion serves as a preventive health strategy [13,14], and there are numerous efforts to promote PA in children and adolescents to develop more active lifestyles [15,16,17,18]. To succeed with the health benefits of PA, it is essential to fully understand the PA determinants. Most research thus far has focused on the psychosociological, social, and environmental issues that affect PA levels [19]. In contrast, the potential effect of intrinsic biological control on regular activity has received little attention [20].

With Rowland’s “ActivityStat” hypothesis [21], the research on biological control that underpins PA and energy expenditure has gained momentum in the literature [22]. Briefly, the ActivityStat hypothesis suggests that an imposed increase or decrease in PA in one domain might induce a compensatory change in the opposite direction in another domain in order to maintain a level of PA or energy expenditure that is overall stable over time [22]. Thus, based on the ActivityStat concept, human beings maintain their total PA at a constant level by adapting various mechanisms, such as increasing or decreasing the frequency, intensity, or duration of time spent engaged in PA [23]. By such adapting mechanisms, their actual energy expenditure can either be increased or decreased so that the overall energy expenditure is stable over a certain period of time. For example, on a day when a child has physical education (PE) classes at school, the child may experience an increase in PA in the morning. However, the child may subsequently increase the time that they spend being sedentary in the afternoon, resulting in an overall PA level that is not increased for this day.

This compensatory mechanism has been observed in some interventional studies: school-based interventions [24,25,26,27,28,29,30,31] demonstrated small or moderate effects on increasing PA within the school setting, but little to no changes in terms of overall PA levels as a result of compensatory mechanisms employed outside of the school setting. Nevertheless, in order to obtain evidence regarding the general effectiveness of interventions on overall PA levels, a holistic approach is important—one that analyzes different domains and time periods of a day (e.g., physical activities in school and outside of school) in addition to overall PA. Such analyses could provide insights into the potential compensatory mechanisms and rearrangements mentioned in the ActivityStat hypothesis.

### 1.3. Displacement Hypothesis

The original displacement hypothesis postulates a mechanism that opposes the ActivityStat hypothesis, stating that watching television and other sedentary behaviors may displace PA [32]. Different studies have suggested that an increased amount of time spent being sedentary is the primary factor contributing to the current increase in obesity seen in adolescents [33,34,35,36,37]. Due to its inverse relationship, this original hypothesis can also be used to justify the displacement of inactivity with PA. However, little evidence exists that supports this assumption in children and adolescents under 18 years of age [38,39,40]. However, a Cochrane systematic review of school-based PA programs has concluded that there is solid evidence that school-based interventions have a positive impact on the duration of PA, with generally no effects on leisure time PA [16]. The review implies no substantial evidence that compensation is being made for PA imposed through interventions by having the PA decrease in another domain [16].

### 1.4. Previous Reviews Analyzing Compensatory Mechanisms in Children and Adolescents

Little is known about how PA varies in children and adolescents within intervals of one and multiple days [22,41]. Additionally, the question arises as to whether and how inactive or sedentary time can be displaced by PA within one day or over a period of several days in children and adolescents. For children and adults, a systematic review by Gomersall, Rowlands, English, Maher, and Olds [22] found inconclusive results with regard to the ActivityStat hypothesis. This review was limited in its scope because it exclusively considered studies that made explicit reference to compensation. Additionally, in the previous review, both children and adults were assessed, and objective as well as subjective measurement methods were used. Overall, to the best of our knowledge, no review has yet analyzed PA compensation in children and adolescents only within intervals of one and multiple days with objective measurement methods.

### 1.5. Aims of the Present Review

Starting from this state of research, this systematic review aims to provide a synthesis of studies that analyzed changes in overall PA among children and adolescents within various contexts of PA. Specifically, the present review aims: (i) to provide an overview and analysis of studies that examined changes in (overall) PA, assessed with objective measurements, in children and adolescents; and (ii) to identify the displacement of PA or whether compensatory mechanisms following PA increase or decrease in one domain or during a timespan of the day (e.g., at school, in the morning) or in terms of the amount at a specific intensity level (e.g., light, moderate, or vigorous PA).

## 2. Methods

This systematic review was performed and is reported in accordance with the Preferred Reporting Items for Systematic Reviews and Meta-Analyses (PRISMA) guidelines [42].

### 2.1. Eligibility Criteria

Studies were deemed to be eligible if they met the following inclusion criteria and were excluded for the review if one or more exclusion criteria applies (Table 1). 

We decided to focus on objective measurement methods due to the advantages over subjective methods. These are limited to poorer reliability and validity, as well as participants recall bias [44]. Objective measurements are able to directly assess one or more dimensions of PA (intensity, frequency, time and type) and a variety of metrics such as step numbers, minutes or intensity [45].

We included interventional as well as non-interventional studies. Interventional studies offered the opportunity to obtain information about “ActivityStat” by investigating within-subject changes in PA levels in response to an intervention stimulus. Non-interventional studies were included if they investigated PA levels of individuals during different timespans or in different domains over a period of several days.

### 2.2. Search Strategy

The search was performed on 31 March 2020 using the electronic databases PubMed, Scopus, Web of Science, and SportDiscus. In contrast to Gomersall, Rowlands, English, Maher, and Olds [22], the search term in the present review was not limited to the term compensation: the search strategy included using a combination of terms related to children and youth (child* OR youth* OR adolescen* OR boy* OR girl* OR student* OR pupil*), terms related to PA compensation (increas* OR decreas* OR “more activ*” OR improve* OR “less activ*” OR compensat* OR displace* OR change* OR activitystat), terms related to measurement methods (“objective* measure*” OR acceleromet* OR pedomet* OR “heart rate monitor*” OR “doubly labeled water” OR calorimet* OR “direct* observ*”), and outcome-related terms (“physical activ*” OR “energy expenditure”). Two filters were used to refine the results and to obtain the final reference sample for screening. In accordance with the set inclusion criteria, studies published between 2000 and 2020 were selected. The publication type was filtered for journal articles (PubMed: “journal article”; SportDiscus: “academic journal”, “peer reviewed”; Scopus: “articles”; Web of Science: “Article”). In accordance with recommendations for systematic reviews [46], we screened the reference lists and citations of included articles in order to identify additional relevant studies.

### 2.3. Study Selection

Identified references were imported into Endnote X9, a reference management software [47]. Subsequently, the citations were imported into Covidence, a systematic review software [48]. Within this program, all duplicates were removed. This step was followed by a three-step study selection process, comprising (1) title-screening, (2) abstract-screening, and (3) full-text-screening for inclusion criteria by two independent reviewers (F.B. and a trained student assistant). During each step of the screening process, all references that could not conclusively be excluded were kept for further screening in the next step. Disagreements between the two reviewers in relation to final inclusion were resolved through discussion with a third researcher (A.K.R.). 

### 2.4. Data Extraction

The following data were extracted from each article: author(s); country; study design; sample description (number of participants, age, sex); aim/purpose of the study; measurement and instrument of measurement as well as duration of measurement; context of PA, main study results on the relationship between PA in one domain and PA in another domain/overall PA; and statistical indicators for compensation. In particular, this was referred to as the *p*-value for between- and within-group comparisons of PA or energy expenditure levels in one domain/timespan and the following domain/timespan (see also Electronic Appendix A).

Studies were also classified by taking into consideration different settings or contexts in which PA was measured. In relation to the main objectives of the study, six settings were defined (school-based PA, physical education, active commuting to school, daily pattern, sport clubs, and others). All included studies were allocated to one of these categories.

### 2.5. Quality Assessment

The methodological quality of the included studies was rated by two independent reviewers (F.B. and a trained student assistant). To assess the quality of each study, criteria for evaluating primary research articles, developed by Kmet et al. [49], were applied. We decided to use this tool because it is appropriate for a variety of different study designs. The “QualSyst” scoring system is based on existing tools and particularly relies upon the instruments developed by Cho and Bero [50] and Timmer et al. [51] for quantitative studies. A series of 14 items was used to assess quality. These items included questions related to study design, methods of participant selection, random allocation procedure, blinding, outcome measures, sample size, estimate of variance, confounding, reporting of results, and the evidence base for the conclusion. The items were scored depending on the degree to which each criterion was met (“yes” = 2, “partial” = 1, “no” = 0). If an item was not applicable to a particular study design, it was coded with “N/A” and was excluded from the calculation of the summative score. Randomization was only scored in interventional studies. The following equation was applied for estimating quality scores: 28 − (number of N/A × 2). Consequently, 28 was the maximum score that could be obtained for the 14 questions. The risk of bias was evaluated with its summary score (range 0–1), whereas higher scores indicated better methodological quality. 

### 2.6. Synthesis of Results

It was anticipated that the studies included in this systematic review would exhibit a diverse range of research methods (e.g., study design, intervention characteristics, setting, measurements, participant characteristics, and outcome measures). Therefore, it was not appropriate to use meta-analysis to integrate and summarize the included studies. Instead, a narrative synthesis of results was performed. Summary tables describing the detailed characteristics of included studies and the visualization of statistical indicators, describing the probability for compensation related to these characteristics, were provided. The included studies and their findings were grouped according to the study design, sample size, target group, dependent variable, geographic origin, PA context, measurement instrument, duration of intervention (if this information was provided), intervention type, and the timespan in which compensation was measured. Analyses of compensatory mechanisms were performed for different categories, which seemed to be helpful for understanding compensation. In this respect, the aim was to determine which study design, target group, measurement instrument, PA context, timespan, and intervention duration, as well as type, was susceptible to compensation. A study was voted to be “supporting compensation” if the overall PA did not change with respect to different time points, or if it did not differ between intervention group(s) and control group(s), or if there was a significant increase in PA in one domain or timespan and a significant decrease in PA in another domain or timespan. 

## 3. Results

### 3.1. Flow Chart

A total of 5917 potentially relevant articles (or 10,946 including duplicates) were identified through database searches, and their titles and abstracts were screened. In the next step, the full texts of 95 studies were retrieved for in-depth screening. Since 20 articles were excluded due to inappropriate aims of study, statistical analysis, participants, or multiple reasons, a total of 75 articles were identified as eligible and were included in this systematic review. Subsequently, 2 additional relevant publications were identified through backward reference tracking, yielding a total of 77 articles reporting on 74 unique studies included in this systematic review (Figure 1). PA as a Civil Skill Program was published in two studies that focused on LPA [52] and moderate PA (MPA) [53], respectively. The PHASE-Study was also reported in two separate articles. Ridgers et al. [54] focused on the association between daily PA on two following days, whereas Ridgers et al. [55] analyzed the correlations between the amount of sitting, standing, and stepping time within and between days in primary schoolchildren. Furthermore, two articles included in this review were part of the TEACHOUT-Study and investigated general effects of education outside the classroom [56] and effects of education outside the classroom in different domains [57].

### 3.2. Characteristics of Included Studies

A complete data extraction table for each included study can be found in Electronic Appendix A. A synthesis of the characteristics of the included articles is presented in Table 2. The majority of included articles (*n* = 49; 64%) presented non-interventional studies and 36% presented interventional studies. The sample size ranged from 13 [58] to 6916 participants [23], with a mean sample size of 532 participants. The geographical origin of the studies was as follows: *n* = 40 Europe, *n* = 21 North/Mid America, *n* = 10 Australia/New Zealand, and *n* = 6 Asia. Most studies (76%) were published between 2011 and 2020, with the earliest being published in 2000. The main focus of 65 studies (84%) were schoolchildren, whereas 12 (16%) studies focused on preschoolers. There were 4 studies that targeted only girls [23,59,60,61], and 2 only targeted boys [62,63]. Only 10 studies explicitly stated that their aim was to test compensation [23,64,65,66,67,68,69,70,71,72], and of these studies, 2 specifically mentioned the ActivityStat hypothesis [68,71]. There were 63 studies that used accelerometry to objectively assess PA, 13 used pedometers, and 1 used a heart rate monitor. A SenseWear Armband was used in 2 studies, in addition to an accelerometer, to assess energy expenditure [71,73]. Included studies were assigned to one of six contexts/settings: school-based PA in children and adolescents (*n* = 28), active commuting to school (*n* = 9), daily PA pattern (*n* = 15), physical education lessons (*n* = 16), organized sports (*n* = 5), or others (*n* = 4). With respect to interventional studies, the duration of the intervention varied from one week [63,71] to two years [30,52,53,74,75]. The average duration of the interventions was 36 weeks (standard deviation (SD) = 37 weeks). Investigation of 3 intervention types found that 50% (*n* = 14) were educational, 29% (*n* = 8) were environmental, and 21% (*n* = 6) were multicomponent. Regardless of their study design (interventional or non-interventional), most studies (85%) examined changes in PA within a one-day period.

A total of 39 of the 77 studies, representing 16,297 (40%) children and adolescents, reported statistical indicators that emphasize the probability of compensation. From these 39 studies, 5 reported inconsistent findings in relation to compensation within the sample [61,66,74,76,77]. Furthermore, some studies indicated compensation in one interventional group (higher amount of PA during intervention) but not in the other [31,72]—in girls but not in boys [56,64,78,79], or in boys but not in girls [80]. In Table 2, an overview of all studies supporting compensation (*n* = 39) is presented. With respect to study design, interventional studies indicated compensatory behavior in 75% of studies (representing 76% of participants; *n* = 6477), whereas non-interventional studies supported compensation in only 30% of studies (representing 30% of participants; *n* = 9820) [54,55,64,66,68,73,77,78,79,80,81,82,83,84,85,86,87,88]. Furthermore, 42% of preschoolers and 40% of schoolchildren showed indicators for compensatory behavior. Measurements conducted with a pedometer indicated compensation in 6938 participants (72%), whereas accelerometer measurements showed indicators for compensation in 9309 participants (30%). In relation to the PA context, school-based PA revealed compensation indicators in 5350 participants, which corresponded to 72% of all participants in this context, followed by active commuting to school (6095 participants; 56%). No compensation was indicated in the sport club context (0%). With respect to intervention duration, in studies with a duration ≥1 year, 100% of participants indicated compensatory behavior. Compensation was supported in all multicomponent interventions (*n* = 2074 participants) and in 64% of all educational interventions (*n* = 2295 participants).

**Table 2 children-09-00351-t002:** Characteristics and compensation or displacement voting of all included articles.

Characteristics	*n* Studies (% of All Studies)	*n* Participants (% of All Participants)	*n* Studies to Support Compensation	*n* Participants to Support Compensation	*n* Studies to Support Displacement	*n* Participants to Support Displacement	All Sources	Compensation Sources
Study design								
Non-interventional studies	49 (64%)	32,310 (79%)	18 (37%)	9820 (30%)	35 (71%)	22,490 (70%)	[23,54,55,58,60,62,64,66,67,68,69,70,73,77,78,79,80,81,82,83,84,85,86,87,88,89,90,91,92,93,94,95,96,97,98,99,100,101,102,103,104,105,106,107,108,109,110,111,112]	[54,55,64,66,68,73,77,79,80,81,82,83,84,85,86,87,88]
Interventional studies	28 (36%)	8519 (21%)	21 (75%)	6477 (76%)	10 (36%)	2042 (24%)	[30,31,52,53,56,57,59,61,63,65,71,72,74,75,76,113,114,115,116,117,118,119,120,121,122,123,124,125]	[30,31,52,53,56,59,61,63,65,72,74,75,114,116,118,121,122,123,124,125]
Sample Size								
<500	57 (74%)	10,980 (27%)					[30,52,53,54,55,56,58,59,60,62,63,64,65,67,68,70,71,72,73,74,76,77,78,79,80,82,83,86,87,88,90,91,92,93,95,96,97,98,99,101,103,105,108,110,111,113,114,115,116,117,118,119,120,122,123,124]	
>500	20 (26%)	29,849 (73)					[23,31,57,61,66,69,75,81,84,85,89,94,100,102,104,107,109,112,121,125]	
Target group								
Preschool children (3–6 years)	12 (16%)	2188 (5%)	5 (42%)	919 (42%)	7 (58%)	38 (58%)	[65,96,103,110,111,113,115,116,117,122,123,124]	[65,116,122,123,124]
School children (>6 years)	65 (84%)	38,641 (95%)	34 (52%)	15,378 (40%)	1269 (58%)	23,263 (60%)	[23,30,31,52,53,54,55,56,57,58,59,60,61,62,63,64,66,67,68,69,70,71,72,73,74,75,76,77,78,79,80,81,82,83,84,85,86,87,88,89,90,91,92,93,94,95,97,98,99,100,101,102,104,105,106,107,108,109,112,114,118,119,120,121,125]	[30,31,52,53,54,55,56,59,61,63,64,66,68,72,73,74,75,76,77,78,79,80,81,82,83,84,85,86,87,88,114,118,121,125]
Dependent Variable								
Physical Activity	76 (96%)	40,779 (99%)					[23,30,52,53,54,55,56,57,58,59,60,61,62,63,64,65,66,67,68,69,70,71,72,73,74,75,76,77,78,79,80,81,82,83,84,85,86,88,89,90,91,92,93,94,95,96,97,98,99,100,101,102,103,104,105,106,107,108,109,110,111,112,113,114,115,116,117,118,119,120,121,122,123,124,125]	
Energy Expenditure	3 (4%)	333 (1%)					[71,73,87]	
Geographic origin								
Europe	40 (52%)	16,881 (41%)					[30,52,53,56,57,58,60,61,68,74,75,77,78,80,82,83,84,85,86,87,88,89,93,94,96,97,99,100,101,102,104,105,106,107,110,111,118,120,121,122]	
North America	21 (27%)	14,256 (35%)					[23,31,59,64,65,66,67,69,70,72,79,90,103,112,113,114,115,116,117,123,124]	
Australia/New Zealand	10 (13%)	3548 (9%)					[54,55,63,71,73,95,98,108,109,125]	
Asia	6 (8%)	6144 (15%)					[62,76,81,91,92,119]	
PA measurement instrument								
Accelerometer	63 (79%)	31,150 (76%)	32 (51%)	9309 (30%)	36 (57%)	21,841 (70%)	[23,30,52,53,54,55,56,57,58,59,60,61,62,63,64,65,66,67,68,69,70,71,72,73,74,75,76,77,78,79,80,81,82,83,84,85,86,88,89,90,91,92,93,94,95,96,97,98,99,100,101,102,103,104,105,106,107,108,109,110,111,112,113,114,115,116,117,118,119,120,121,122,123,124,125]	[30,52,53,54,55,56,59,61,63,64,65,66,68,73,74,75,77,78,80,82,83,84,86,88,114,116,118,121,122,123,124,125]
Pedometer	13 (16%)	9629 (23%)	6 (46%)	6938 (72%)	10 (77%)	2691 (28%)	[31,58,70,72,76,79,81,85,90,92,93,109,120]	[31,72,76,79,81,85]
Heart Rate Monitor	1 (1%)	50 (0.1%)	1 (100%)	50 (100%)	0	0	[87]	[87]
Context of PA measures								
School-based PA	26 (34%)	7392 (18%)	19 (72%)	5350 (72%)	10 (36%)	2042 (28%)	[30,31,52,53,56,57,59,61,63,65,71,72,74,75,76,113,114,115,116,117,119,120,122,123,124,125]	[30,31,52,53,56,59,61,63,65,72,74,75,76,114,116,122,123,124,125]
Active commuting to school	10 (13%)	10,733 (26%)	5 (50%)	6175 (58%)	7 (70%)	4558 (43%)	[78,80,81,83,95,104,107,108,109,118]	[78,80,81,83,118]
Daily PA Pattern	15 (19%)	12,499 (31%)	7 (47%)	1677 (13%)	8 (53%)	10,822 (87%)	[23,54,55,67,69,73,77,82,84,87,92,93,96,103,112]	[54,55,73,77,82,84,87]
Physical Education	17 (22%)	5814 (14%)	7 (41%)	2567 (44%)	12 (75%)	3247 (56%)	[62,64,68,79,85,86,88,90,91,97,99,100,106,110,111,121]	[64,68,79,85,86,88,121]
Sports Club	5 (6%)	2583 (6%)	0 (0%)	0 (0%)	5 (100%)	2583 (100%)	[58,60,94,98,101]	
Others (Locations/Active play)	4 (5%)	1808 (4%)	1 (25%)	528 (29%)	3 (75%)	1280 (71%)	[66,89,102,105]	[66]
Duration of Intervention								
≤one week	2 (7%)	207 (2%)	1 (50%)	51 (25%)	1 (50%)	156 (75%)	[63,71]	[63]
≤one month	1 (4%)	67 (1%)	0 (0%)	0 (0%)	1 (100%)	67 (100%)	[117]	
≤1–2 months	6 (21%)	869 (12%)	5 (83)	573 (66%)	2 (33%)	296 (34%)	[72,76,114,120,122,123]	[72,76,114,122,123]
≤3–4 months	3 (11%)	144 (2%)	3 (100%)	144 (100%)	0	0	[59,65,118]	[59,65,118]
≤5–6 months	6 (21%)	2241 (38%)	3 (50%)	1520 (68%)	4 (67%)	721 (32%)	[31,113,115,116,119,125]	[31,116,125]
≤one year	4 (14%)	2450 (28%)	3 (75%)	1648 (58%)	2 (50%)	802 (32%)	[56,57,121,124]	[56,121,124]
>one year	6 (21%)	2541 (31%)	6 (100%)	2541 (100%)	0	0	[30,52,53,61,74,75]	[30,53,61,74,75,85]
Intervention type								
Educational	14 (50%)	3593 (42%)	9 (64%)	2295 (64%)	7 (50%)	1145 (32%)	[56,57,61,63,71,72,113,114,115,116,117,122,123]	[56,59,61,63,114,116,122,123,125]
Environmental	8 (29%)	2393 (28%)	6 (75%)	2108 (88%)	2 (25%)	285 (12%)	[52,53,65,75,118,119,120,121]	[52,53,65,75,118,121]
Multicomponent	6 (21%)	2533 (30%)	6 (100%)	2074 (82%)	1 (17%)	459 (18%)	[30,31,74,76,124,125]	[30,31,74,76,124,125]
Timespan measuring compensation								
Within a day	65 (85%)	37,198 (91%)	35 (54%)	15,567 (42%)	36 (55%)	21,631 (58%)	[23,30,31,52,53,56,57,58,59,61,62,63,65,66,67,68,69,71,72,74,75,76,77,78,79,80,81,82,83,84,85,86,87,88,89,91,92,94,95,96,100,102,103,104,105,106,107,108,109,110,111,112,113,114,115,116,117,118,119,120,121,122,123,124,125]	[30,31,52,53,56,59,61,63,65,66,68,74,75,76,77,78,79,80,81,82,83,84,85,86,87,88,114,116,118,121,122,123,124,125]
Between two consecutive days	4 (5%)	904 (2%)	3 (75%)	610 (67%)	1 (25%)	294 (33%)	[54,55,73,97]	[54,55,73]
Across several days	8 (10%)	2727 (7%)	1 (13%)	120 (4%)	8 (100%)	2607 (96%)	[60,64,70,90,93,98,99,101]	[64]

### 3.3. Results of Methodological Quality Assessment

For quality assessment, we applied the “QualSyst” scoring system with a scoring range of 0–1 [49]. The mean quality score of the included articles was rated as high by both raters (mean = 0.87, SD = 0.10, range 0.5–1). The methodological quality criteria and the proportion of studies fulfilling the criteria are presented in Table 3; more detailed quality assessments of each included article are presented in Electronic Appendix A. Items 5, 6, and 7 were only scored for randomized controlled trials (RCTs).

In RCTs, the mean quality score was 0.76 (SD = 0.11) (range 0.5–0.89). In all RCTs, poor blinding of the treatment was evident for both investigators and participants. Overall, the included studies revealed a high quality, and only few studies exhibited a higher risk of bias [64,65,86,120].

## 4. Discussion

The present systematic review aims to provide a synthesis of studies that have analyzed changes in overall PA, assessed using objective measurements, or compensatory behavior caused by PA increases or decreases in a specific PA domain or during the timespan of one day in children and adolescents.

A total of 77 articles were included that investigated compensation or displacement across various contexts in children and adolescents. Overall, approximately 50% of the included articles found indicators suggesting compensation, and 50% refuted compensational behavior and supported the displacement of inactive time with bouts of activity. Detailed analyses based on study design, target group, instruments, context, intervention duration, and measurement duration were performed, revealing differences in compensation depending on categories. The analyses showed tendencies toward compensation in school interventions (especially with durations lasting longer than 1 year) and tendencies of displacement in the context of weekly organized participation in sport clubs.

It is hypothesized that when PA in one domain or timespan increases, PA in another domain or timespan decreases in order to maintain the constant PA level, as postulated by the ActivityStat hypothesis [22]. In the present analysis, 38 of 77 articles, involving a total of 24,532 participants, refuted the ActivityStat hypothesis and showed an increase in overall PA that resulted from imposed PA in one domain or timespan and absence of a reduction in PA in other domains or timespans. Sustained displacement of inactivity with PA led to an overall increased PA level, as described in the displacement hypothesis. One possible explanation for these increases in overall PA could be that imposed PA stimuli serve as some kind of trigger: PA opportunities in different contexts may stimulate children and adolescents to engage more in physical activities during the entire day [64,70,78,91,105,107]. 

In our review, we distinguished between interventional and non-interventional studies. A total of 21 interventional studies (*n* = 6477 participants) showed indicators for compensation, whereas only 18 non-interventional studies (*n* = 9820 participants) indicated compensatory behavior. This suggests that when the PA of children and adolescents is promoted in an intervention, the participants tend to compensate for the additional bouts of PA within the intervention by decreasing their activity levels during other parts of the day or in other domains so that they maintain their overall PA at a stable level. On the other hand, when children increase their PA levels on their own, without participating in an intervention program (in non-interventional studies), it seems that they do not compensate for this, and ultimately increase their overall PA level. This could be due to the fact that, in these cases, their PA is more likely based on intrinsic motivation that external influences. Furthermore, interventions were mostly offered and performed in (pre)school contexts. Improvements in (pre)school PA can be compensated for by less PA outside of (pre)school [30,31,52,53,56,59,61,63,74,75,76,114,116,118,121,122,123,124,125]. “It is possible that school-based interventions are too focused on school setting and children and adolescents do not translate the health message on the importance of physical activity at home or in the community” [16]. For school interventions, it has been suggested that the focus should also be placed on changing parental behaviors and awareness for the sake of adopting a sustainable active lifestyle. In addition, multicomponent interventions or interventions that include schools together with families or communities are most effective in changing PA levels [16,29]. In general, interventions are most efficient when they operate on multiple levels [126]. “According to ecological models, the most powerful interventions should (a) ensure safe, attractive, and convenient places for physical activity, (b) implement motivational and educational programs to encourage use of those places, and (c) use mass media and community organization to change social norms and culture.” [127]. Intrapersonal, interpersonal, organizational, community, and public policy factors can influence health behaviors; thus, they consequently counteract compensatory behavior. Even though the literature suggests that multicomponent interventions have been shown to be useful for changing PA behavior, our results contradict this assumption, with the findings of compensatory behavior in all multicomponent studies [30,31,74,76,124,125].

Analysis of interventions pointed out a wide range in terms of intervention duration. All six studies (*n* = 2541 participants) in which interventions lasted for over one year supported compensation behavior in children and adolescents [30,52,53,61,74,75]. Nevertheless, compensatory behavior in children and adolescents was also identified in interventions that had a duration between one month and one year [31,59,65,72,76,114,116,118,122,123,125]. A possible explanation for this finding could be that interventions which last for a shorter period of time may have little or no effect in changing the PA behavior of children and adolescents due to a lack of time needed to progress through the six stages of change, according to the trans-theoretical model [128]. This means that children and adolescents who accumulate more MVPA during an intervention might continue to be as physically active in their leisure time as they were before the intervention. The longer an intervention lasts, the greater the probability that children and adolescents adapt their PA and become less active in their leisure time; hence, maintaining their overall PA at a stable level.

Almost all studies included in our review captured PA data using pedometers or accelerometers. Only 6 pedometer studies (*n* = 6938 children and adolescents) reported compensatory mechanisms [31,72,76,79,81,85], whereas children and adolescents (*n* = 9309) showed compensatory behavior in 32 accelerometer studies [30,52,53,54,55,56,59,61,63,64,65,66,68,73,74,75,77,78,80,82,83,84,86,88,114,116,118,121,122,123,124,125]. One explanation for this finding could be that pedometers only capture step counts—an index of the number of steps a person took—whereas the overall PA levels for participating individuals remained unknown. Hence, it is likely that compensation is diagnosed through a measured reduction in steps, whereas other shifts in PA levels (e.g., overall MVPA) remain unconsidered. Furthermore, only 1 study out of the 77 analyzed studies investigated energy expenditure using a heart rate monitor and indicated compensation [87].

In addition to interventions in a school context, there are two other settings in which children and adolescents are physically active. Only 57% of children and adolescents who actively commute to school showed indicators for compensation [78,80,81,83]. Active commuting and the independent mobility of children provide additional opportunities for spontaneous play [107] and enable other active behaviors [129,130]. This can lead to an increase in overall PA, which therefore supports the displacement theory. 

In the PE context, only 32% of the participants indicated compensatory behavior [64,68,79,85,86,88]. PE classes should provide an opportunity for children and adolescents to engage in PA and to develop knowledge about and attitudes toward developing an active lifestyle [131], which could lead to displacing inactivity with active behavior. Interestingly, two articles, involving 365 participants, investigated the impact of different amounts of PE per week on overall PA levels in children and adolescents. From their findings, it can be summarized that more PE per week is not necessarily effective for increasing the total PA, because the PA in PE classes is often compensated for by less activity outside of the school setting [68,86]. Consequently, future studies should assess what the right amounts and intensities of PA during PE classes would be in order to avoid compensation outside of school.

Finally, our detailed analyses revealed one PA domain in which an increase in activity levels was not found to lead to compensation, but instead, to displacement: when engaging in organized sport clubs, children and adolescents do not compensate their PA levels by being less active after the training sessions [58,60,94,98,101]. Sport clubs represent a health-promoting setting and support children and adolescents in living an active lifestyle outside sport clubs [132]. Furthermore, sport programs can provide beneficial access to and resources for recreational activities [98]. Thus, participation in sport clubs serves as an additional factor for increasing overall PA and can displace sedentary behavior.

Compensatory behavior occurs after a PA increase or decrease in one domain or timespan in order to maintain a stable overall PA level. Almost all studies in this review revealed that a PA increase in one domain or timespan is followed by a PA decrease in another domain or timespan, which is negatively connoted. Nevertheless, there exists one study [79] in our review, where compensatory behavior was found to occur after a PA reduction—leading to compensation being positively connoted.

### 4.1. Implications

This review of compensation for PA in children and adolescents provides inconsistent results relating to compensation. Consequently, further research is needed to better understand compensatory mechanisms, and a recommendation is made for future studies to investigate PA behavior over a period of a few days using an objective measurement method. In addition, participants should complete a questionnaire or keep a diary in order to terminate and locate their activities and to obtain information about the reasons for their PA behavior. Social support plays an important role for sufficient PA in children and adolescents. Thus, PA behavior and attitudes of family and friends can influence one’s own PA and determine compensatory behavior. Additional subgroup analysis, including an examination of differences in PA by gender, age, weight status, socio-economic status (SES), and ethnicity, could provide more information about compensatory behavior. Gender differences have already been seen in a few of the included studies with inconclusive results [56,64,78,80]. Additionally, various SES analyses indicate different environmental, social, and educational circumstances [133,134]. Hence, SES is an important predictor of PA in children and adolescents [133] and can influence compensatory behavior. Unfortunately, none of the included studies investigated compensatory behavior separately for different SES. It is hypothesized that children and adolescents with lower SES compensate more often than individuals with higher SES. It would also be interesting to further investigate the setpoint for “ActivityStat” or possible differences depending on age, season, or energy intake. Through an experimental design, future studies could investigate when this setpoint is reached and whether there are differences. Furthermore, there are currently no existing theories that deal with the timeframe for compensation. It is hypothesized that the timeframe for compensation is unlikely to be day-to-day [22]. Currently, the timeframes in the studies examined in our review are random. Finally, combined measurements of energy expenditure and PA should be used to obtain more detailed and reliable information about compensatory mechanisms.

Practical implications refer to interventional studies: in addition to active PA promotion, it is important to improve awareness in children and adolescents, as well as in their parents, regarding the importance of PA, as well as to encourage them to be physically active at home during their leisure time. This is necessary in order to avoid compensation that occurs when PA at home and/or in the family environment is reduced after increases in PA levels take place during interventions in, for example, the school setting.

### 4.2. Strengths and Limitations

The main strength of this review is that we exclusively included studies that objectively measured PA, including measures that directly assessed one or more PA dimension (e.g., frequency, intensity, time, and type) and captured a variety of measures, such as step counts, activity minutes, and PA intensity [45]. An additional strength lies in the fact that the systematic search of relevant primary studies employed several electronic databases and a comprehensive list of search strings. Furthermore, the reference lists of all included studies were manually checked in the search for additional relevant studies. Our search strategy was broad enough to enable us to identify relevant studies as well as to include those studies that did not analyze PA compensation as their main objective. In contrast to Gomersall, Rowlands, English, Maher, and Olds [22], we did not only include studies that made explicit reference to compensation. Instead, we analyzed studies investigating changes in overall PA and in different domains or time segments for compensatory mechanisms. Another strength is the inclusion of a wide range of different settings in which PA plays an important role. 

A limitation of this review relates to the variety in the study designs of the included studies, which made a comparison of the results difficult. Additionally, some studies only allowed between-subject analyses, which, in turn, only enabled conclusions about compensation to be obtained from a comparison of PA levels between two groups. For better understanding of compensatory mechanisms, within-subject analyses could provide stronger results. Another limitation is that there were different PA segments in the reviewed studies, which made it difficult to compare them all. 

## 5. Conclusions

This systematic review provided inconclusive results regarding potential compensatory activity behavior after changes in PA levels in one domain or during a timespan in children and adolescents. Overall, 39 studies (*n* = 16,297 children and adolescents) that were included in this review did exhibit indicators of compensation. In summary, the synthesis of the included studies revealed a tendency for compensatory behavior in the context of interventions, especially in interventions with a long duration (<1 year). Furthermore, children and adolescents who regularly participated in organized sports showed no indicators for compensatory behavior. In order to verify the results of the present review, further investigations are needed.

## Figures and Tables

**Figure 1 children-09-00351-f001:**
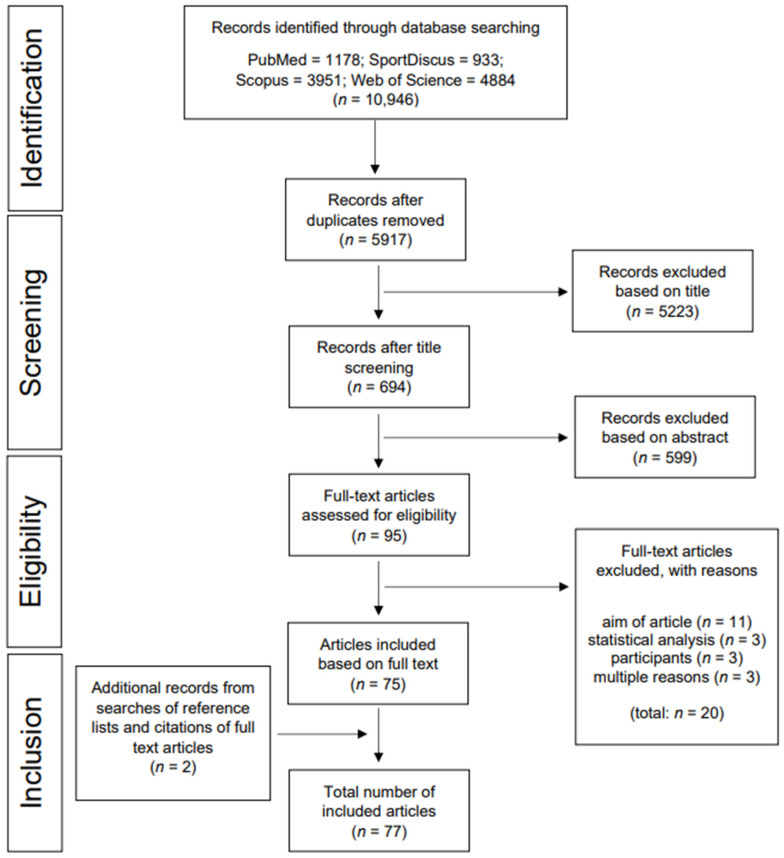
Flow chart.

**Table 1 children-09-00351-t001:** Inclusion and exclusion criteria for the systematic review.

	Inclusion Criteria	Exclusion Criteria
Measurement method	Objectively measured PA or energy expenditure	Subjectively measured dependent or independent variables
Outcome	Investigation and analysis of changes in (overall) PA or (overall) energy expenditure caused by an increase in PA in a specific domain or during a specific timespan (e.g., at school; in the morning)	Studies with only one measurement point; Studies that do not report an overall PA or energy expenditure score.
Population	Participants in the study were healthy; Studies with children and adolescents (3 to 19 years of age, or their mean age was in this range) [43].	Participants with chronic diseases; Studies among participants with a competitive athletic background; Participants under the age of 3 or older than 19 years.
Publication type	Studies published in a peer-reviewed journal in English or German	Grey literature; Publications without peer review; Publication language other than German or English.
Publication date	Articles published in the year 2000 or later	Articles published before 2000

**Table 3 children-09-00351-t003:** Criteria for the methodological quality assessment and the number (%) of studies that scored points or each criterion.

		Studies Fulfilling the Criteria *n* (%)			
No.	Item	Yes	Partial	No	N/A
1	Question/objective is sufficiently described?	51 (66%)	26 (34%)	0	0
2	Study design is evident and appropriate?	75 (97%)	2 (3%)	0	0
3	Method of subject/comparison group selection or source of information/input variables is described and appropriate?	64 (83%)	13 (17%)	0	0
4	Subject (and comparison group, if applicable) characteristics are sufficiently described?	50 (65%)	27 (35%)	0	0
5	If interventional and random allocation was possible, was it described?	7 (9%)	9 (12%)	1 (1%)	60 (78%)
6	If interventional and blinding of investigators was possible, was it reported?	4 (5%)	1 (1%)	12 (16%)	60 (78%)
7	If interventional and blinding of subjects was possible, was it reported?	0	1 (1%)	16 (21%)	60 (78%)
8	Outcome(s) and (if applicable) exposure measure(s) is/are well defined and robust to measurement/misclassification bias? Means of assessment are reported?	66 (86%)	11 (14%)	0	0
9	Sample size is appropriate?	70 (91%)	7 (9%)	0	0
10	Analytic methods are described/justified and appropriate?	75 (97%)	2 (3%)	0	0
11	Is some estimate of variance reported for the main results?	36 (47%)	41 (53%)	0	0
12	Controlled for confounding?	44 (57%)	31 (40%)	2 (3%)	0
13	Results reported in sufficient detail?	58 (75%)	19 (25%)	0	0
14	Conclusions supported by the results?	58 (75%)	19 (25%)	0	0

## Data Availability

Not applicable.

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
