# Peer review of "Compensation or Displacement of Physical Activity in Children and Adolescents: A Systematic Review of Empirical Studies"

_children, 2022, doi:10.3390/children9030351_

Round 1
Reviewer 1 Report
This is a systematic review of the literature on physical activity (PA) and/or energy expenditure (EE) in children and adolescents (3–19 yrs), analyzing behavior patterns of PA/EE in time to support or refute the ActivityStat hypothesis. This is somewhat like a proof-of-concept study.
In general, I found this manuscript a little bit dense. Although it contains interesting information, I think Introduction section could be simplified, particularly, but not limited to, sub-sections 1.1. and 1.2. Methods are clear except for what the authors designate “statistical indicators for compensation.” This must be clearer so that we can understand how have authors more objectively decided on compensation vs. displacement. I suspect the p-value in group comparisons, but this must be reported.
Lines 17–18: I think this is better written in the past tense.
Line 22: I think this should be reworded because the assessment of compensatory mechanisms was performed by you not the primary studies. Remember that this was as important argument you have used to differentiate this review from the one conducted by Gomersall, Rowlands, English, Maher and Olds [27] (lines 478–479).
Line 137: I don’t understand this sentence. Can the authors please explain or reword?
Line 159: I believe “exist” may be a typo…
Line 162: When stating “Intra-individual” do you mean “within-subject”?
Line 196: Please provide example(s) of what are “statistical indicators for compensation”.
Line 302: “intervention” instead of “invention”?
Line 487: “could provide” instead of “provided”?
Line 491: I think “Based on Rowland [26]” may be removed. It does not add anything to the conclusion of this SR.
Author Response
Please see the attachement.

Reviewer 2 Report
Congratulations to the authors for their rigorous work, well thought out in its objectives and methodology.
I would just like a few minor questions to be answered.
Line 121-122 states "little evidence exists that supports this assumption in children and adolescents under 18 years of age [46]". However, the quote is from 2004. I think that to make that assertion the quote should be more recent. The same is true for quote 47.
In the conclusion line 500 the reference to "Based on Rowland [26]" is not very clear. I would ask for an attempt to explain it a little more clearly.
Author Response
Please see the attachement.

Round 2
Reviewer 1 Report
I'm satisfied with the amendments the authors have performed in the revised version of the manuscript. I believe the quality and readability of the article have significantly improved. I have a minor suggestion in line 219 though. In the portion of the sentence "...the p-value for group and within-subject comparisons…", I think it would be clearer to readers if you write instead "...the p-value for between- and within-group comparisons…".
Nice work. Best wishes.